# Precise Modeling of Thermal and Strain Rate Effect on the Hardening Behavior of SiC/Al Composite

**DOI:** 10.3390/ma15062000

**Published:** 2022-03-08

**Authors:** Yanju Wang, Pengfei Wu, Xiaolei He, Wei Zhao, Xiang Lan, Yanshan Lou

**Affiliations:** 1Materials Evaluation Center for Aeronautical Aeroengine Applications, AECC Beijing Institute of Aeronautical Materials, Beijing 100095, China; lanxiang_buaa@163.com; 2School of Mechanical Engineering, Xi’an Jiao Tong University, Xi’an 710049, China; wpf901229@stu.xjtu.edu.cn (P.W.); 3120301248@stu.xjtu.edu.cn (W.Z.); 3Institute of Aluminum Alloys, AECC Beijing Institute of Aeronautical Materials, Beijing 100095, China; okhere007@126.com

**Keywords:** SiC/Al composite, hardening behavior, coupling effect of temperature and strain rate, constitutive modeling

## Abstract

Temperature and strain rate have significant effects on the mechanical behavior of SiC/Al 2009 composites. This research aimed to precisely model the thermal and strain rate effect on the strain hardening behavior of SiC/Al composite using the artificial neural network (ANN). The mechanical behavior of SiC/Al 2009 composites in the temperature range of 298–623 K under the strain rate of 0.001–0.1 s^−1^ was investigated by a uniaxial tension experiment. Four conventional models were adopted to characterize the plastic flow behavior in relation to temperature, strain rate, and strain. The ANN model was also applied to characterize the flow behavior of the composite at different strain rates and temperatures. Experimental results showed that the plastic deformation behavior of SiC/Al 2009 composite possesses a coupling effect of strain, strain rate, and temperature. Comparing the prediction error of these models, all four conventional models could not provide satisfactory modeling of flow curves at different strain rates and temperatures. Compared to the four conventional models, the suggested ANN structure dramatically improved the prediction accuracy of the flow curves at different strain rates and temperatures by reducing the prediction error to a maximum of 4.0%. Therefore, the ANN model is recommended for precise modeling of the thermal and strain rate effect on the flow curves of SiC/Al composites.

## 1. Introduction

Ceramic-particle-reinforced metal matrix composites combine the characteristics of easy deformation and high thermal conductivity of metals with the characteristics of high strength and low expansion of ceramic particles [1]. They are widely applied in the fields of aerospace, automobile, and electronic packaging [2,3]. As one of the metal matrix composites, silicon carbide (SiC)-reinforced aluminum matrix (SiC/Al) composites possess high strength, high modulus, and enhanced temperature resistance. However, the addition of SiC easily causes complicated deformation behavior of composites. Moreover, the high hardness and brittleness of reinforced SiC limits plastic flow of the matrix alloy, resulting in poor machinability [4]. At the same time, the loading condition has a significant effect on the mechanical response of materials. Plastic deformation mechanisms have been extensively studied for metals and metallic-based composites [5,6,7]. Mechanical experiments and numerical characterization are the key methods to investigate the deformation process of SiC/Al composites.

The temperature and strain rate are two important factors that influence flow stress during the plastic deformation process of SiC/Al composites. The material behavior can be described by the mathematical model. An appropriate constitutive model is a prerequisite for accurate numerical analysis of material deformation. So far, a large number of constitutive models have been proposed or improved to characterize the dynamic characteristics of materials under different conditions. In general, these constitutive models are divided into phenomenological and physical-based types according to features of the models. Among the phenomenological models, the Johnson–Cook (JC) equation [8] is the most popular due to the simplicity of form and ease of application. Khan and Huang [9] and Huh et al. [10] considered the coupling effect of strain and strain rate and developed dynamic hardening models for a wide range of strain rates. Song and Huh [11] proposed an improved model (MKH) by modifying the strain rate hardening term in the K–H model. Piao et al. [12] developed a modified L–H (MLH) model by considering the coupling effect of temperature and strain rate. As for physical-based models, the classical Zerilli–Armstrong (ZA) model was proposed by Zerilli and Armstrong [13] on the basis of microstructure evolution characteristics. Based on the analysis of physical properties and microstructure of materials during deformation, other models have been developed by Khan and Liu [14], Djordjevic et al. [15], Chen et al. [16], and Gupta and Bronkhorst [17]. The above models describe the plastic flow behavior of specific materials under corresponding loading condition with high accuracy. However, these models have limitations more or less for other materials, such as SPRC340S steel, OFHC copper, and Ti6Al4V alloy.

In recent years, the deformation behavior of SiC/Al composites has been investigated by many researchers and characterized by constitutive models. Tirtom et al. [18] characterized the strain-rate-dependent compression mechanical behavior of an SiC-particulate-reinforced Al (2024-O) metal matrix composite (MMC) with different particle volume fractions. Based on the processing map, Rajamuthamilselvan et al. [19] researched the effect of temperature on the plastic behavior and microstructure evolution of 20%SiC/Al 7075 composite. Moćko et al. [20] carried out experimental investigation on SiC/Al composites with 0%, 10%, 20%, and 30% SiC with a wide range of strain rates and large magnitudes of strains and modeled the deformation behavior. Hao and Xie [21] described the flow behavior of 30%SiC/Al 2009 composite at 350–500 °C under strain rates of 0.01–10 s^−1^ using a Zener–Hollomon parameter in the hyperbolic sine form, with the results indicating that the flow stress is positively correlated with strain rate and negatively correlated with temperature. Yuan et al. [22] evaluated suitability of the modified Zerilli–Armstrong model, the strain compensation Arrhenius-type model, the double multivariate nonlinear regression (DMNR) model, and the artificial neural model (ANN) for the hot deformation behavior of SiC/Al composites. Zhang et al. [23] developed a 3D-microstructure-based finite element model to simulate the elastoplastic response of a 7%SiC/Al composite under the condition of room temperature and strain rate of 5 × 10^−4^ s^−1^. Based on a representative volume element, Zhang et al. [24] adopted the microscopic finite element model to accurately characterize the deformation behavior of 17%SiC/Al 2009 composite. Tang et al. [25] established a modified JC constitutive model to characterize the coupling effect of strain, temperature, and strain rate of 17vol%SiC/Al 7055 composite. Although there have been many studies on SiC-reinforced composites, the strain rate and thermal effects have rarely been investigated for these composites, especially for the newly developed 15%SiC/Al 2009 composite.

Accordingly, this study aimed to investigate the effect of temperature and strain rate on the stress–strain response of 15%SiC/Al 2009 composite. Mechanical experiment was carried out under uniaxial tension. The temperature range was 25–350 °C, and the strain included 0.001, 0.01, and 0.1 s^−1^. The JC, ZA, MKH, and MLH models were adopted to characterize material behavior. The artificial neural network (ANN) was also applied to characterize the flow curves at different strain rates and temperatures. The suitability of these models was evaluated by comparing their predictions with the experimental results.

## 2. Experimental Procedures

### 2.1. Experimental Material

The parent material in this study was 15%SiC/Al 2009 composite extruded bar. The matrix was Al 2009, whose chemical composition is listed in Table 1. α-SiC with an average diameter of 8 µm was used as the reinforcing particle. The volume fraction was 15% for SiC particles. The composite material was prepared by powder metallurgy. The process flow was as follows: first, Al 2009 powder was prepared by supersonic gas atomization; then, a finely mixed SiC and Al 2009 powder was loaded into the package by hot iso-static compression after vacuum degassing; finally, the hot isostatic pressing ingot was hot extruded on a 3600 t extruder with an extrusion ratio of 9:1. The microstructure was observed by a Leica DM4000 optical microscopic machine, as shown in Figure 1. It was observed that the SiC particles were distributed more randomly and homogeneously in the Al2009 matrix on the LT–ST plane compared to the L–LT plane. However, on closer view, the particle distribution was not as homogeneous and random as expected. The composite could be approximately viewed as a homogeneous continuum from a macroscopic point of view because the particle size was much smaller than the specimen size for the mechanical experiment, as shown in Figure 2. Moreover, the composite material was densely structured with approximately zero porosity because no obvious voids were observed. The particle size was about 8 µm, and the specimen width was about 750 times the particle diameter. Therefore, the mechanical behavior was measured by mechanical tensile tests at different strain rates and temperatures based on the assumption that the material was homogeneous and uniform.

### 2.2. Mechanical Test

The two-dimensional structure of a uniaxial tension specimen is shown in Figure 2. The parallel gauge length was 30 mm for the design specimens. The experimental specimen was cut along the longitudinal direction of the parent material. The thickness of the specimens was 2.4 mm. The range of strain rates and temperatures were 0.001–0.1 s^−1^ and 298–623 K, respectively. Each experiment was repeated 3–5 times under the same loading condition of strain rate and temperature. All experiments were carried out on the ETM504C universal testing machine. The surface of each specimen was sprayed with uniform black and white speckle. Apart from the experiment at 298 K, other specimens were heated in the environment box. Before the tension loading, specimens were kept evenly for about one minute at 373–623 K before being slowly heated to the specified temperature. The deformation process of each specimen was recorded by the XTOP digital image correlation (DIC) system. About 200 pictures with resolution of 2448 × 2048 were collected for each experiment. The force was measured by a load cell. The measured force was transmitted to the DIC system. The DIC system recorded the images of deformed specimens and the force simultaneously at an identical rate. The stroke ∆L was measured by a visual extensor with an initial gauge length of 15 mm located at the center of the parallel gauge length to calculate the engineering stress σ and strain ε.

## 3. Result

The experimental results under different temperatures and strain rates are shown in Figure 3. The initial gauge was 15 mm along the longitudinal direction. It should be noted that the true strain in Figure 3 includes both the elastic and plastic strain, but the elastic strain has been removed from the total strain in the figures thereafter. The true stress–true plastic strain curves were used for calculation, parameter calibration, and evaluation in this study. It was found that the true stress gradually decreased with increasing temperature, while the stroke before fracture approximately increased with temperature at each strain rate, especially when the testing temperature was higher than 423 K. This was due to the thermal softening effect at high temperature. Moreover, the variation of true stress–true strain curves with temperature for 15%SiC/Al 2009 composite was different from that for Al 2009. The reason for this was the addition of SiC with high strength. The decline of force for each true stress–true strain curve was due to the occurrence of necking behavior. The deformation behavior before the maximum force belonged to uniform deformation, which was used to compute the flow curves.

Based on the assumption of homogeneous deformation, the true stress–true plastic strain relationship can be calculated before the maximum force as follows:(1)$$\left\{ \begin{array}{l} \sigma =F/A\\ \varepsilon =\Delta L/L_{0}\\ \sigma_{true}=\sigma (1+\varepsilon )\\ \varepsilon_{true}=\ln(1+\varepsilon )\\ \varepsilon_{p}=\varepsilon_{true}-\sigma_{true}/E \end{array}\right.$$
where *A* and *L*_0_ are the initial cross-sectional area and gauge, respectively. *σ_true_*, *ε_true_*, and *ε_p_* denote the true stress, true strain, and true plastic strain, respectively. The analytical calculation is not correct for inhomogeneous deformation after necking at the maximum force. The stress–strain curve at large strains after necking should therefore be determined by other advanced methods, such as the inverse engineering approach [26,27]. The computed stress–strain curves are compared in Figure 4 and Figure 5 at different temperatures and strain rates. As can be seen, the curves mostly decreased with temperature, indicating the thermal softening effect for the composite. The thermal softening effect was more obvious at high temperature, especially when the temperature was higher than 423 K. Moreover, the elongation was mostly higher for higher temperature experiments, as shown in Figure 3. However, the necking strain was not always higher at higher temperatures, as shown in Figure 4 and Figure 5. This can also be observed by the strain distribution at the beginning of necking or the maximum force in Figure 6 for each strain rate and temperature. The strain distribution at the necking was very consistent with the results in Figure 4 and Figure 5 in relation to the necking strain at different temperatures and strain rates.

Figure 4 shows the effect of temperature on the true stress–true plastic strain curves at three different strain rates, and Figure 5 presents the effect of strain rates on the plastic behavior of 15%SiC/Al 2009 composite at seven different temperatures. The plastic deformation clearly showed a variation in nonuniformity as the temperature increased at each strain rate. The phenomenon uncovers a coupling effect of temperature and strain on the strength of 15%SiC/Al 2009 composite. As can be seen, the plastic strength of 15%SiC/Al 2009 composite gradually increased with the strain rate when the strain rate increased from 0.01 to 0.1/s, but the strain rate effect was not apparent from 0.001 to 0.01/s at temperatures less than 423 K. At high temperature, a strong positive strain rate effect was observed for the strain rate range studied in this research. At 298 K, a slightly negative strain rate effect was observed when the strain rate increased from 0.001 to 0.01/s at a smaller strain of less than about 0.01, but the strain rate effect changed to positive at larger strain higher than 0.03, as shown in Figure 5a. At this temperature, the strain rate effect was observed to be strongly positive for higher strain rate from 0.01 to 0.1/s. Moreover, there was strong thermal softening effect, especially for high temperatures and high strain rates. The result reflects the positive correlation between the strain rate and the plastic flow strength in 15%SiC/Al 2009 composite. However, there were some differences in the effect of strain rate on the plastic strength under different temperatures. This reveals the coupling effect of strain rate and temperature. Therefore, the plastic deformation behavior of 15%SiC/Al 2009 composite under different temperatures and strain rates possesses a coupling effect of strain, strain rate, and temperature.

## 4. Constitutive Modeling

### 4.1. JC Model

On the basis of mechanical experiment, Johnson and Cook [8] proposed a plastic flow model describing the effect of strain, strain rate, and temperature on the strength of metals as follows:(2)$$\sigma \left(\varepsilon ,\dot{\varepsilon },T\right)=\left[\sigma_{r}\left(\varepsilon \right)\right]\left[\sigma_{\dot{\varepsilon }}\left(\dot{\varepsilon }\right)\right]\left[\sigma_{T}\left(T\right)\right]=\left[A+B{\left(\varepsilon \right)}^{n}\right]\left[1+Cln\frac{\dot{\varepsilon }}{{\dot{\varepsilon }}_{0}}\right]\left[1-{\left(\frac{T-T_{r}}{T_{m}-T_{r}}\right)}^{m}\right]$$
where *σ* and *ε* are the equivalent stress and equivalent plastic strain, respectively; $\dot{\varepsilon }$ and $\dot{\varepsilon }/{\dot{\varepsilon }}_{0}$ denote the strain rate and dimensionless plastic strain rate (${\dot{\varepsilon }}_{0}$ is the reference strain rate which is usually 1 s^−1^), respectively; and *T*, *T_r_*, and *T_m_* represent the absolute temperature, reference temperature, and metal melting temperature, respectively. *A*, *B*, *n*, *C*, and *m* are five material constants, where *A* is the initial yield stress, *B* is the hardening constant, *n* is the hardening exponent, *C* is the strain rate constant, and *m* is the thermal softening exponent.

Because the melting point of aluminum is lower than that of SiC, the values of *T_m_* was set to the melting temperature of the aluminum alloy of 923 K. Room temperature was adopted as the value of *T_r_*, namely 298 K. Based on the experimental true stress–plastic strain curve presented in Figure 4 and Figure 5, the material constants of the JC model were calibrated by adopting the least square method with the Levenberg–Marquardt algorithm, as listed in Table 2. The predicted stress–strain relationship, shown in Figure 7, was compared with the experimental results. Results showed that the predicted result of the JC model presented an obvious proportional change with temperature. The JC model could only well predict the deformation behavior under limited conditions even though the effect of strain rate and temperature was reasonably predicted. A big difference was observed for several loading conditions, such as at 623 K with 0.001 s^−1^.

### 4.2. ZA Model

Zerilli and Armstrong [13] described the effect of temperature and strain rate on the flow stress of body-centered cubic structure (BCC) and face-centered cubic (FCC) metals based on the dislocation mechanism. The ZA constitutive model is expressed as follows:(3)$$\sigma =C_{0}+\left(C_{1}+C_{2}\sqrt{\varepsilon }\right)\exp(-C_{3}T+C_{4}Tln\dot{\varepsilon })+C_{5}\varepsilon^{n}$$
where *C*_1_, *C*_2_, *C*_3_, *C*_4_, and *C*_5_ are the material constants. Equation (3) is the hardening model of BCC metals when the value of *C*_2_ is equal to 0. It is transformed into the plastic flow equation of FCC metals when *C*_1_ = *C*_5_ = 0. *C*_0_ is the initial yield stress at room temperature and the reference strain rate that is related to the microstructure, *C*_2_ is the hardening constant, *C*_3_ is the thermal softening constant, and *C*_4_ is the coupling constant between temperature and strain rate.

Because the Al 2009 matrix takes up most of the volume in 15%SiC/Al 2009 composites, the volume fraction is low (~15%) for the particles. Moreover, plastic deformation is caused by the slip mechanism of the Al 2009 matrix, and the deformation of particles is elastic. Therefore, it is reasonable to treat the plastic behavior of the composite as a kind of aluminum alloy. Therefore, in this study, 15%SiC/Al 2009 composite was viewed as an aluminum alloy, and the ZA FCC model was adopted to describe the effect of strain rate and temperature on the flow curves of the composite. Based on the least square method, the material constants of the ZA model were calibrated, which are summarized in Table 3. The predicted and experimental results are presented in Figure 8. Compared to the predicted result of the JC model, the ZA model presented a coupling effect of temperature, strain, and strain rate. However, there was a big difference between the predicted stress–stain relationship and the experimental data. The prediction accuracy of the ZA model did not improve the prediction accuracy compared to the JC model even though the ZA model had the ability to model the coupled effect of strain rate and temperature.

### 4.3. MKH Model

Song and Huh [11] developed a modified K–H model by combining the advantages of the K–H and JC models in order to improve the prediction accuracy of the plastic flow behavior of metals. The constitutive equation is written as follows:(4)$$\sigma =\left[A+B{\left(1-\frac{ln\dot{\varepsilon }}{lnD_{0}^{p}}\right)}^{n_{1}}\varepsilon^{n_{0}}\right]\left[1+C{\left(ln\frac{\dot{\varepsilon }}{{\dot{\varepsilon }}_{0}}\right)}^{p}\right]\left[1-{\left(\frac{T-T_{r}}{T-T_{m}}\right)}^{m}\right]$$
where the value of $D_{0}^{p}$ is the reference strain rate for coupled strain and strain rate effect, which is recommended to be equal to $10^{9}$ s^−1^; *A* is the initial yield stress at the reference strain rate and temperature; *B* is the hardening constant; *n*_1_ and *n*_0_ are the hardening exponents, respectively; *C* is the strain rate constant; *p* is the strain rate exponent; and *m* is thermal softening exponent.

On the basis of the experimental data illustrated in Figure 4 and Figure 5, the material constants of Equation (4) were calibrated by the least square method, as listed in Table 4. The calibrated stress–strain relationship is shown in Figure 9. Comparing the predicted and experimental results, it was found that the MKH model showed better predicted result for the deformation behavior of 15%SiC/Al 2009 composite in relation to temperature and strain rate than the JC and ZA models for most of the loading conditions. However, the discrepancy between the MKH prediction and experiments was still high for the strain rates of 0.01 and 0.1 s^−1^. This indicates that the coupled MKH model can partially consider the effect of coupled strain, strain rate, and temperature on flow curves, but its predictability is limited and needs to be enhanced.

### 4.4. MLH Model

Huh et al. [10] summarized the experimental results of 4340 steel, OFHC copper, and Ti6Al4V alloy under various strain rates and compared the prediction accuracy of different strain rate hardening models. Piao et al. [12] proposed the MLH model by introducing the thermal softening term to the L–H model, as follows:(5)$$\sigma \left(\varepsilon ,\dot{\varepsilon },T\right)=A{\left(\varepsilon +\varepsilon_{0}\right)}^{n}\left[\frac{1+q\left(\varepsilon \right){\dot{\varepsilon }}^{p\left(\varepsilon \right)}}{1+q\left(\varepsilon \right){\dot{\varepsilon }}_{r}{}^{p\left(\varepsilon \right)}}\right]\left[1-{\left(\frac{T-T_{r}}{T_{m}-T_{r}}\right)}^{m\left(\dot{\varepsilon }\right)}\right]$$
with
(6a)$$q\left(\varepsilon \right)=\frac{q_{1}}{{\left(\varepsilon +q_{2}\right)}^{q_{3}}}$$
(6b)$$p\left(\varepsilon \right)=\frac{p_{1}}{{\left(\varepsilon +p_{2}\right)}^{p_{3}}}$$
(6c)$$m\left(\dot{\varepsilon }\right)=m_{1}+m_{2}\ln\left(\frac{\dot{\varepsilon }}{{\dot{\varepsilon }}_{r}}\right)$$

The first item in the MLH model represents the strain hardening effect, where *A*, *ε*_0_, and *n* are strain hardening coefficients. The second term denotes the strain rate effect under the specified strain conditions, where *q*_1_, *q*_2_, *q*_3_, *p*_1_, *p*_2_, and *p*_3_ are the strain rate hardening coefficients. The last term is the temperature softening term that is capable of coupling the strain rate effect, where *m*_1_ and *m*_2_ are the temperature softening coefficients.

The material constants of the MLH model were calibrated based on the least square method, as summarized in Table 5. The predicted and experimental results were compared, as shown in Figure 10. As can be seen, the predicted result of the MLH model was very similar to that of the MKH model and characterized the effect of temperature and strain rate on the deformation behavior of 15%SiC/Al 2009 composite with better agreement compared to other models. As indicated in Equation (5), the MLH model can couple the strain, strain rate, and temperature effect on the flow curves with enhanced predictability. This is the reason the predicted flow curves matched the experimental measurement with better accuracy than the JC, ZA, FCC, and MKH models. However, there were apparent differences between the prediction and experiments, as shown in Figure 10. Therefore, further modification of the current model or advanced approaches are required to precisely consider the strain rate and temperature effect.

## 5. ANN Model

Comparing the predicted flow curves and experimental results, it was found that the four conventional models above could not model the flow curves of the composite at different strain rates and temperatures. Recently, the ANN model has increasingly been adopted to model the highly nonlinear plastic behavior of metals, such as DP steel [28] and AA5182-O [29]. In this research, the back propagation (BP) ANN model was adopted to characterize the flow curves of the composite with the coupled effect of strain rate and temperature. Generally, the accuracy of the BP model can be improved by increasing the number of hidden layers and the neuron number of each hidden layer. However, the numerical computation efficiency severely decreases with the parameter number and the complexity of the BP network. After comprehensive analysis and parameter study of the BP network, a single hidden layer BP network was suggested as a proper balance between accuracy and numerical computation efficiency. Therefore, the chosen BP network was 3-16-1, as shown in Figure 11. There were three layers in the BP network: the input layer with three neurons denoting the three inputs of strain, strain rate, and temperature; one hidden layer with 16 neurons; and one output layer with the stress as the output. Therefore, the total parameter number in the BP network was 3×16+16+16×1+1=81. The activation function was set as tansig from the input layer to the hidden layer and purelin from the hidden layer to the output layer.

A total of 90% of the experimental data was randomly selected as the training set, while the rest 10% was used at the testing set to validate the trained BP model. The BP network was trained by the training set in Figure 4. With the trained BP network, the flow curves were predicted at different strain rates and temperatures. Different temperatures are compared in Figure 12a for 0.001 s^−1^, Figure 13a for 0.01 s^−1^, and Figure 14a for 0.1 s^−1^. The comparison showed no apparent difference between prediction by the trained BP model and the experimental results at different strain rates and temperatures. To further investigate the prediction accuracy of the BP model, the errors were calculated for the predicted flow curves at different strain rates and temperatures. The computed errors are shown in Figure 12b for 0.001 s^−1^, Figure 13b for 0.01 s^−1^, and Figure 14b for 0.1 s^−1^. At the strain rates of 0.001 and 0.01 s^−1^, the maximum error was observed at the beginning of plastic deformation, which was less than 2.5%. At 0.1 s^−1^, the largest error was observed at the onset of plastic deformation, which was about 4.0% for this strain rate. For most of the plastic strain, the difference between the BP prediction and experiments was less than 1.0%, which is very good compared to the four conventional models above. All the comparisons above show that the BP network can precisely characterize the thermal effect on the flow curves at different temperatures and strain rates.

## 6. Evaluation of Four Conventional Models and the BP Network

From the comparison of the flow curves predicted by the four conventional models and the experimental results in Figure 7, Figure 8, Figure 9 and Figure 10, it is obvious that the JC, ZA, MKH, and MLH models could not precisely model the flow curves at all the strains, strain rates, and temperatures investigated in this research. The BP prediction and experiments compared in Figure 12, Figure 13 and Figure 14 at different strain rates and temperatures show than the BP network was very high in accuracy compared to the four conventional model. To comprehensively evaluate the four conventional models of JC, ZA, MKH, and MLH and the BP network regarding their prediction accuracy of the flow curves at different strain rates and temperatures, the *R*^2^ (coefficient of determination) and the average absolute relative error were are calculated by the 21 experimental curves in Figure 4 and Figure 5 using Equation (7). It should be noted that all the experimental data were used to compute *R*^2^ and error for the four conventional models and the BP model. Moreover, *R*^2^ and error were calculated by the training and testing sets, respectively, for the BP model. The calculated *R*^2^ for the BP model was 0.99986 and 0.99866 for the training and testing sets, respectively. The error predicted by the BP model was 4.56892 × 10^−4^ for the training set and 1.75169 × 10^−3^ for the testing set. For the BP model, it was observed that *R*^2^ and the error computed for the testing set were larger than those for the training set. However, the difference was not big compared to the computed *R*^2^ and the error for the four conventional models. Therefore, *R*^2^ and error of the BP model computed by all the experimental data were used for comparison with the four conventional models in Figure 15.
(7)$$\left\{ \begin{array}{l} R^{2}=1-\frac{\sum_{i=1}^{n}{\left(stress_{pred\_i}-stress_{exp\_i}\right)}^{2}}{\sum_{i=1}^{n}{\left(stress_{exp\_i}-\bar{stress_{exp}}\right)}^{2}}\\ error=\frac{1}{n}\sum_{i=1}^{n}\left|\frac{Stress_{pred\_i}-Stress_{\exp\_i}}{Stress_{\exp\_i}}\right|\times 100 \end{array}\right.$$
where *n* is the number of all experimental data presented in Figure 4 and Figure 5, and $\bar{stress_{exp}}$, *Stress_pred_i_*, and *Stress*_exp*_i*_ are the average value of experimental true stress, predicted true stress, and experimental true stress, respectively. The calculated *R*^2^ and average absolute relative error of the JC, ZA, MKH, and MLH models and the BP model are shown in Figure 15a for *R*^2^ and Figure 15b for the average absolute relative error. As can be seen from the comparison, the order of the prediction accuracy for the above four models was BP > MLH > MKH > JC > ZA. Among the four conventional models, the MLH model provided better accuracy than the rest of the three models, but the *R*^2^ was just slightly higher than 0.8, which is not satisfactory. The *R*^2^ of the BP model reached 0.99974, which indicates that the BP model characterized the effect of strain rate and temperature on flow curves with highly promising reliability. The result indicates that the BP model is more suitable for characterizing the effect of temperature and strain rate on the deformation behavior of 15%SiC/Al 2009 composite than the four conventional MLH, MKH, ZA, and JC models. Moreover, the average absolute relative error of the BP model was 5.86372 × 10^−4^, which is less than 1.0 × 10^−4^ times the four conventional models. This is the reason the average absolute relative error bar of the ANN model cannot be observed in Figure 15b, which also demonstrates the higher prediction accuracy of the BP model compared to the four conventional models. The results indicate that the BP network could dramatically reduce the prediction error of the flow curves at different strain rates and temperatures for the composite. The coefficient number of the 3-16-1 ANN model was about 10 times that of the JC, ZA, MKH, and MLH models. However, considering the dramatic improvement of prediction accuracy of the ANN model, it is worth increasing the coefficient number by ANN to improve the prediction accuracy of the flow curves at different strain rates and temperatures considering that all the traditional models could not satisfactorily predict the effect of strain rate and temperature on the flow curves for the composite.

The predicted stress–strain curves are given in Figure 7 for JC, Figure 8 for ZA, Figure 9 for MKH, Figure 10 for MLH, and Figure 12, Figure 13 and Figure 14 for the BP model. It is apparent that the predicted yield stress at zero plastic strain for the ZA model was identical at different strain rates and temperatures, but the other three models predicted different yield stresses at this condition. This is because *C*_1_ = *C*_5_ = 0 was assumed for the ZA FCC model. However, the experimental results showed that the yield stress at zero plastic strain was strongly affected by the strain rate and temperature. Therefore, the ZA FCC model could not model the strain rate and temperature effect on the yield stress at zero plastic strain. Based on comparison of the predicted and measured flow curves, all the four conventional models adopted in this study could not precisely describe the effect of the strain rate and temperature on the strain hardening behavior of the composite. The ANN-based BP model was the best according to the prediction accuracy of the flow curves for the composite considering the effect of strain rate and temperature. Moreover, the numerical computation cost using the BP model did not increase dramatically compared to the conventional model as an identical BP structure was adopted in this study according to [29]. Lastly, the prediction accuracy of the trained ANN model is guaranteed only in the range of 25–350 °C and 0.001–0.1 s^−1^, so its predictability out of this range is under question, especially when the loading condition is far off from the range. To ensure the prediction accuracy of a special loading condition, it is suggested that the ANN model be trained by experimental results conducted in conditions around the special loading condition.

Among the four conventional models, the ZA FCC, MKH, and MLH models completely or partially coupled the strain, strain rate, and thermal effects, while the JC model treated the strain, strain rate, and thermal effects separately. However, the sum square errors of these four models indicate that ZA FCC was the worst compared to the other three models. The best accuracy was obtained by the MLH model. The sum square error of the JC model was slightly higher than the MLH model even though the JC model treated the strain hardening, strain rate hardening, and thermal softening separately and did not consider their coupling effects at all. Considering the simplicity and relatively high accuracy of the JC model compared to the MKH and MLH models, the JC model is very competitive for the modeling of strain rate and thermal effect of 15%SiC/Al 2009 composite in numerical analysis and design of hot forging processes. The strong coupling effect of the strain rate and temperature on the flow curves of 15%SiC/Al 2009 composite is a big challenge for analytical modeling by conventional models. However, the prediction accuracy can be dramatically enhanced if the BP model is adopted to characterize the flow behavior at different strain rates and temperatures.

## 7. Conclusions

In this research, the deformation behavior of 15%SiC/Al 2009 composite under various temperatures and strain rates was investigated by tensile tests. The predicted results of the JC, ZA, MKH, and MLH models and the BP network were compared with the experimental true stress–plastic strain curves. The suitability of the four conventional models and the BP network was evaluated by the sum square errors. The conclusions are as follows:(1)The experimental result indicates that there is a strong correlation between the strain rate and the plastic flow strength of 15%SiC/Al 2009 composite. The temperature has a remarkable influence on the plastic flow behavior of 15%SiC/Al 2009 composite. The plastic deformation behavior of 15%SiC/Al 2009 composite under different temperatures and strain rates possesses strong coupling effect of strain, strain rate, and temperature.(2)The true stress–plastic strain curves under different temperatures and strain rates were modeled by the four popular strain rate hardening and thermal softening models of JC, ZA, MKH, and MLH. The characterized deformation behavior of the MLH model had better agreement with the experimental data than that of the JC, ZA, and MKH models as the MLH model is capable of modeling the coupling effect of strain, strain rate, and temperature. However, all the conventional models failed to characterize the flow curves at different strain rates and temperatures with satisfactory accuracy for the composite.(3)Compared to the conventional models, the ANN-based BP model was shown to highly improve the prediction accuracy of the yield stress at different strains, strain rates, and temperatures for the composite used in this study. Precise modeling of the strain rate and thermal effects for the composite is a great challenge for conventional models, and the BP model is an alternative but very competing approach for accurate modeling of strain rate and thermal effect of the composite. Therefore, the BP model is recommended to characterize the plastic behavior of the composite in numerical simulation and analysis of warm forging and plastic forming.

There is one important limitation of the BP model that needs to be improved, namely the prediction accuracy of the trained BP model is only guaranteed when the composite is deformed under conditions in the range of 25–350 °C and 0.001–0.1 s^−1^ because the model is trained by the experimental data of these loading conditions. However, its predictability beyond the range needs further investigation in future study.

## Figures and Tables

**Figure 1 materials-15-02000-f001:**
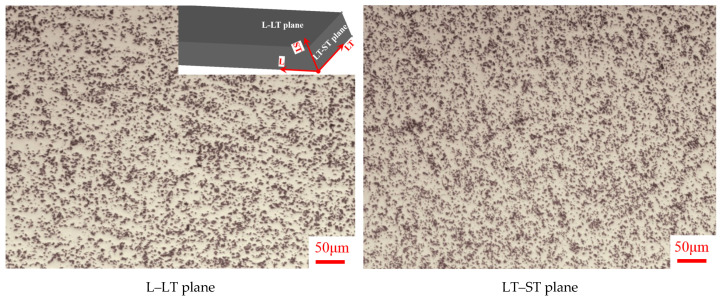
Microstructure of 15%SiC/Al 2009 composite extruded bar under high-power microscope (L: longitudinal, LT: longitudinal–transverse, and ST: short–transverse). The cross-sectional dimension is 50 mm (ST) by 105 mm (LT).

**Figure 2 materials-15-02000-f002:**
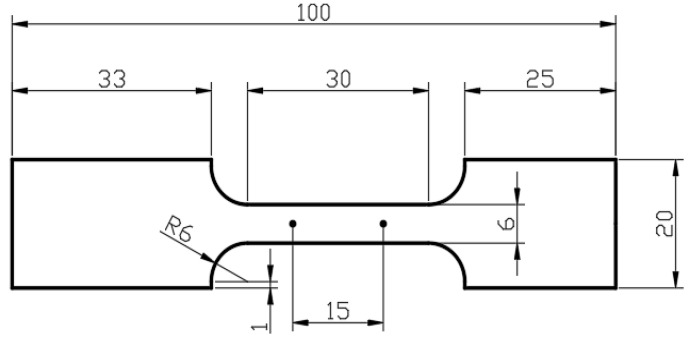
Uniaxial tension specimen (unit: mm).

**Figure 3 materials-15-02000-f003:**
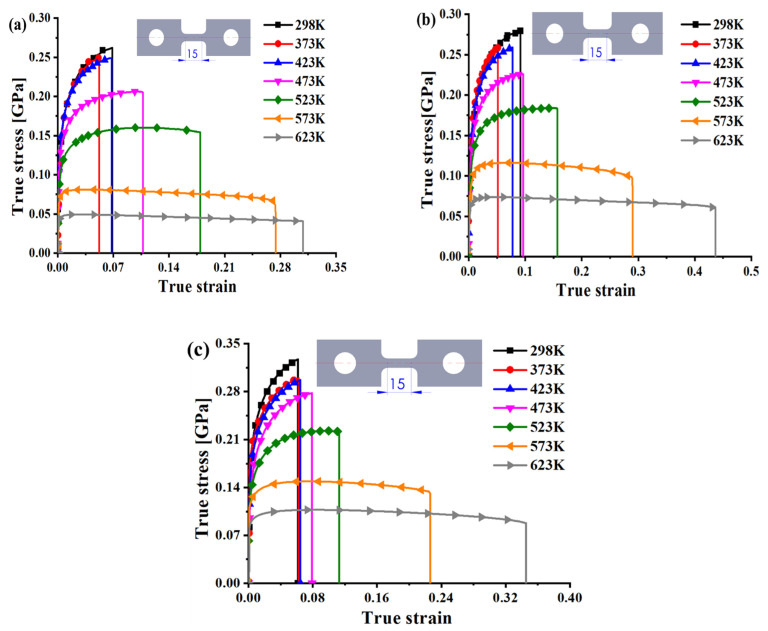
Experimental result of 15%SiC/Al 2009 composite under various strain rates: (**a**) 0.001 s^−1^, (**b**) 0.01 s^−1^, and (**c**) 0.1 s^−1^.

**Figure 4 materials-15-02000-f004:**
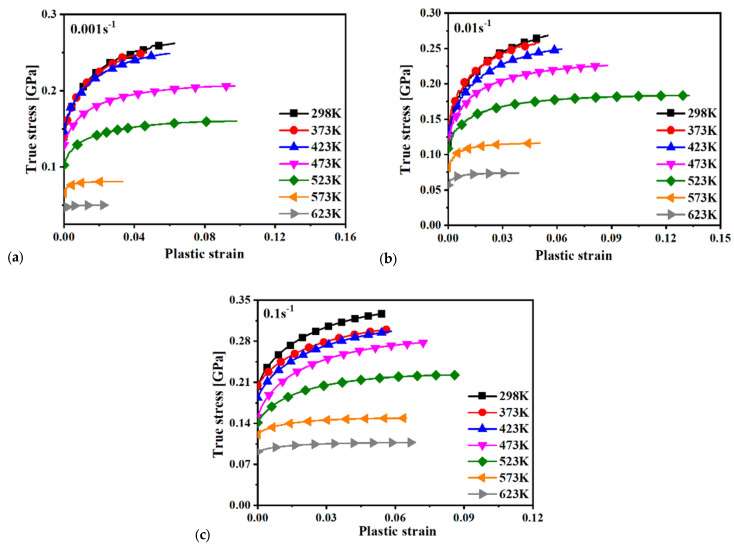
Effect of temperature on the true stress-plastic strain curves of 15%SiC/Al 2009 composite under different strain rates: (**a**) 0.001 s^−1^, (**b**) 0.01 s^−1^, and (**c**) 0.1 s^−1^.

**Figure 5 materials-15-02000-f005:**
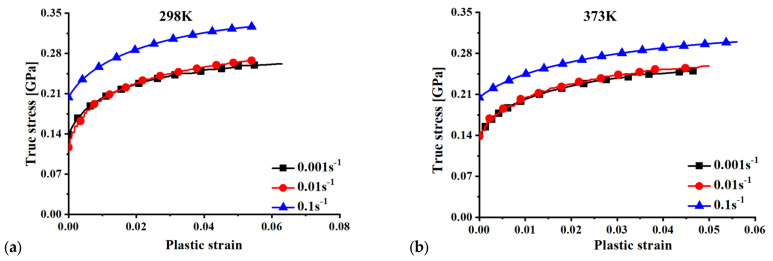
Effect of strain rates on the true stress-plastic strain curves of 15%SiC/Al 2009 composite under different temperatures: (**a**) 298 K, (**b**) 373 K, (**c**) 423 K, (**d**) 473 K, (**e**) 523 K, (**f**) 573 K, and (**g**) 623 K.

**Figure 6 materials-15-02000-f006:**
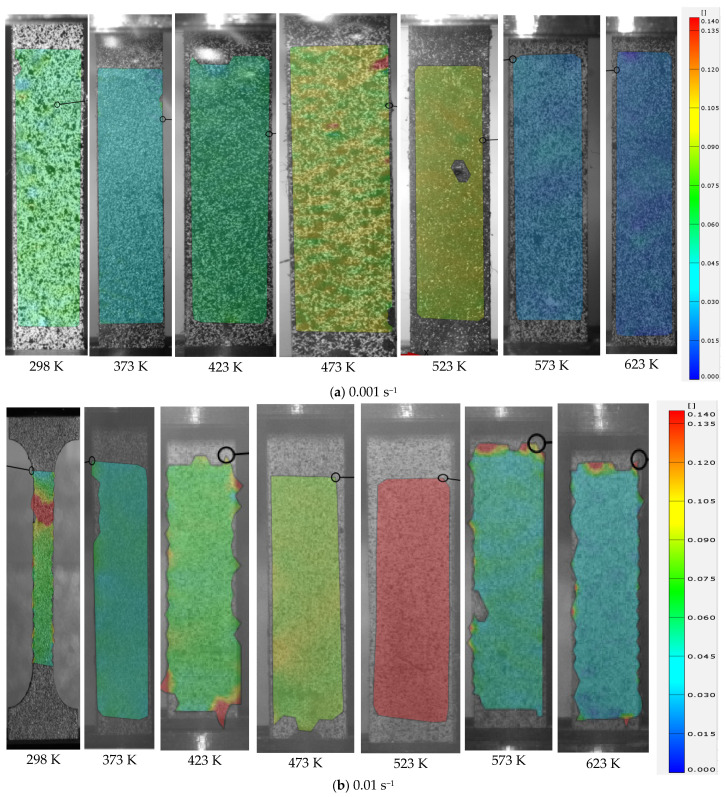
Distribution of equivalent strain measured by DIC at different temperatures for different strain rates: (**a**) 0.001 s^−1^, (**b**) 0.01 s^−1^, and (**c**) 0.1 s^−1^.

**Figure 7 materials-15-02000-f007:**
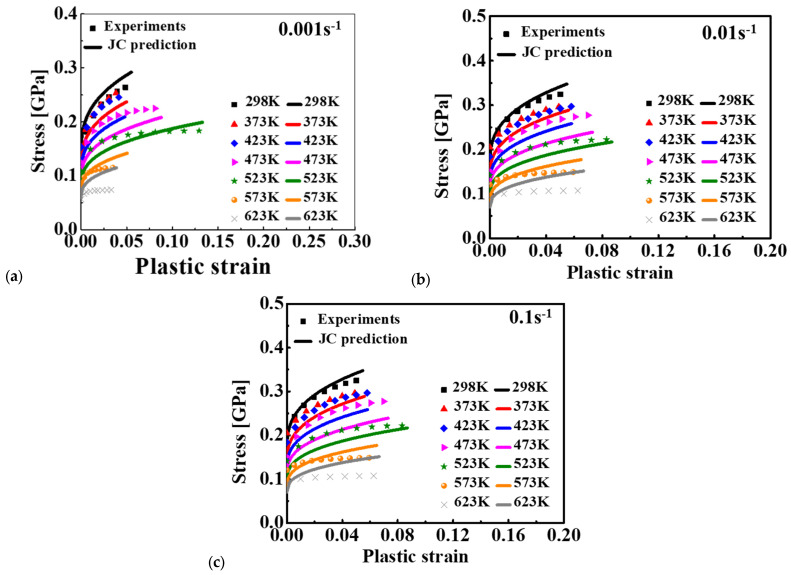
Comparison of the JC stress-strain curves and experimental results for 15%SiC/Al 2009 composite at different strain rates: (**a**) 0.001 s^−1^, (**b**) 0.01 s^−1^, and (**c**) 0.1 s^−1^.

**Figure 8 materials-15-02000-f008:**
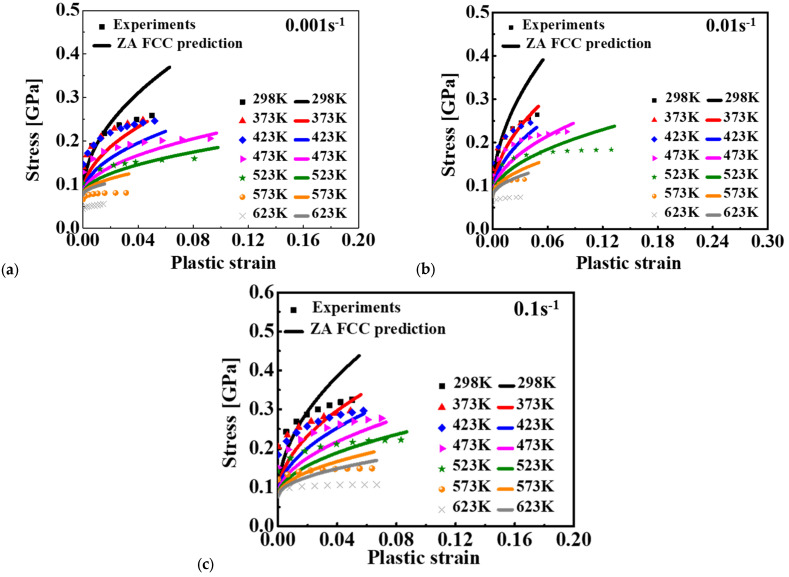
Comparison of the stress-strain curve between the ZA model and experimental results for 15%SiC/Al 2009 composite at different strain rates: (**a**) 0.001 s^−1^, (**b**) 0.01 s^−1^, and (**c**) 0.1 s^−1^.

**Figure 9 materials-15-02000-f009:**
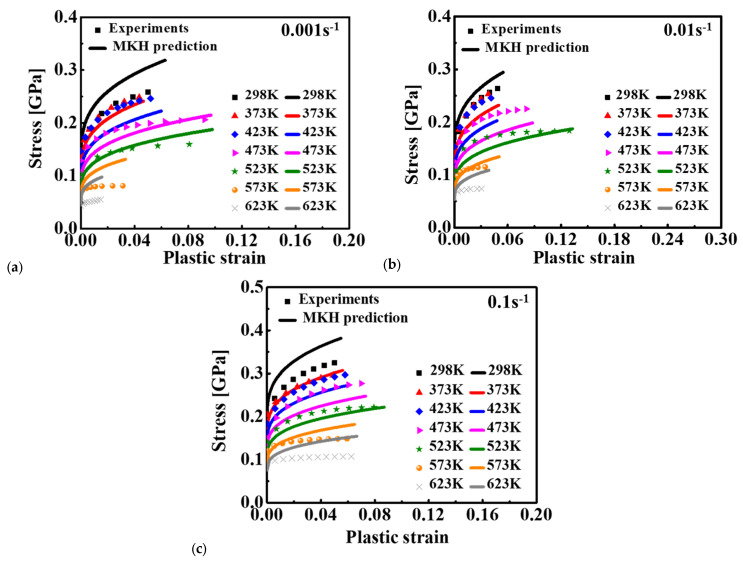
Comparison of the MKH predicted results and experimental data for 15%SiC/Al 2009 composite at different strain rates: (**a**) 0.001 s^−1^, (**b**) 0.01 s^−1^, and (**c**) 0.1 s^−1^.

**Figure 10 materials-15-02000-f010:**
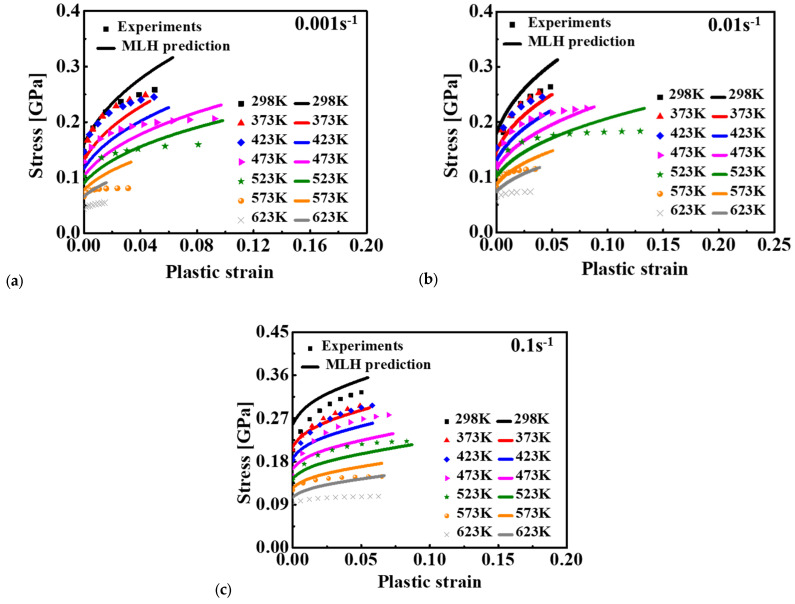
Comparison of the MLH predicted results and experimental results for 15%SiC/Al 2009 composite at different strain rates: (**a**) 0.001 s^−1^, (**b**) 0.01 s^−1^, and (**c**) 0.1 s^−1^.

**Figure 11 materials-15-02000-f011:**
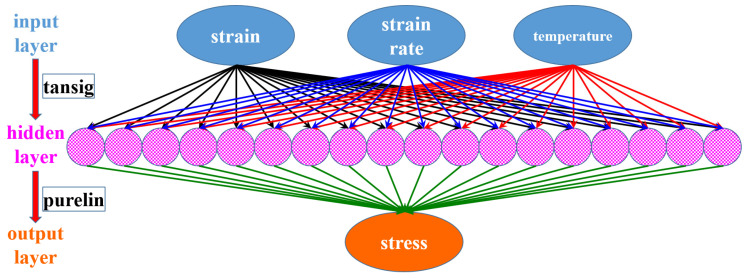
The chosen BP network for the flow curve modeling of the composite.

**Figure 12 materials-15-02000-f012:**
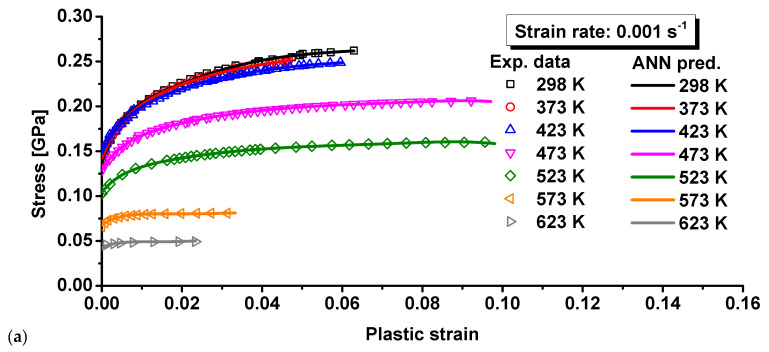
Comparison of the BP predicted results and experimental results for 15%SiC/Al 2009 composite at 0.001 s^−1^: (**a**) the flow curves and (**b**) the prediction error.

**Figure 13 materials-15-02000-f013:**
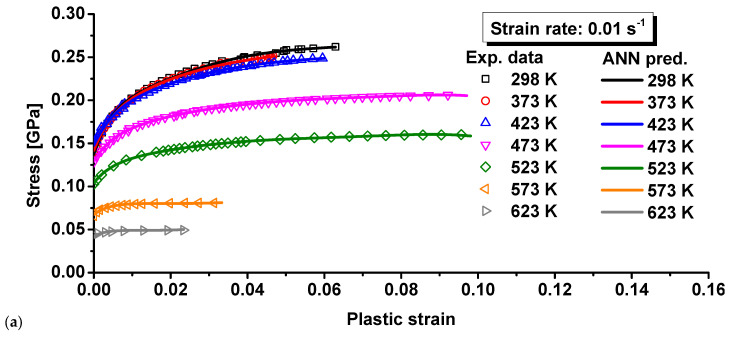
Comparison of the BP predicted results and experimental results for 15%SiC/Al 2009 composite at 0.01 s^−1^: (**a**) the flow curves and (**b**) the prediction error.

**Figure 14 materials-15-02000-f014:**
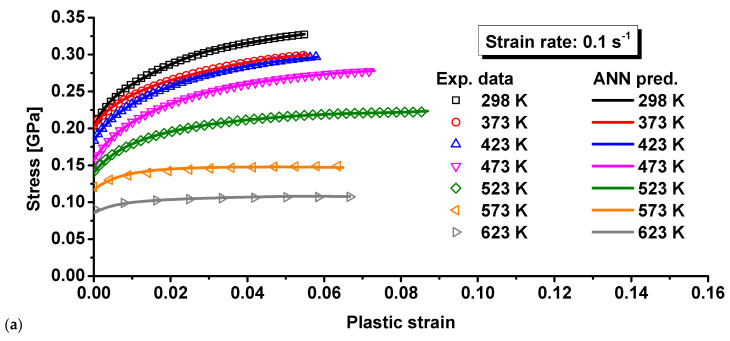
Comparison of the BP predicted results and experimental results for 15%SiC/Al 2009 composite at 0.1 s^−1^: (**a**) the flow curves and (**b**) the prediction error.

**Figure 15 materials-15-02000-f015:**
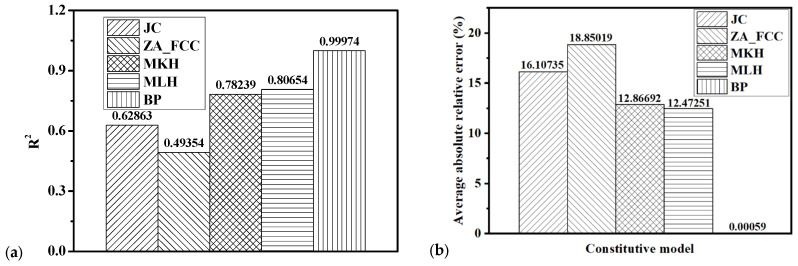
Comparison of (**a**) the *R*^2^ (coefficient of determination) and (**b**) the average absolute relative error for JC, ZA, MKH, and MLH models and the BP network.

**Table 1 materials-15-02000-t001:** The chemical composition of Al2009 alloy powders.

Composition	Cu	Mg	Fe	Si	O	Al
wt %	3.2–4.4	1.0–1.6	<0.1	<0.1	<0.1	Bal.

**Table 2 materials-15-02000-t002:** Material constants of the JC model.

*A*	*B*	*n*	*C*	*m*
0.1896	0.6139	0.3575	0.0463	0.7843

**Table 3 materials-15-02000-t003:** Material constants of the ZA model.

$C_{0}\;$	$C_{2}\;$	$C_{3}\;$	$C_{4}\;$
0.0758	5.8315	0.0040	2.077 × 10^−4^

Note: *C*_1_ = *C*_5_ = 0.

**Table 4 materials-15-02000-t004:** Material constants of the MKH model.

*A*	*B*	*m*	$n_{1}$	$n_{0}$	*C*	*p*
0.1260	0.2679	0.7218	2.0539	0.3148	0.0034	3.2904

**Table 5 materials-15-02000-t005:** Material constants of the MLH model.

*A*	$\varepsilon_{0}$	*n*	$p_{1}$	$p_{2}$	$p_{3}$	$q_{1}$	$q_{2}$	$q_{3}$	$m_{1}$	$m_{2}$
0.5296	0.0014	0.1865	0.2948	0.2864	2.5321	0.0001	0.0386	7.5816	0.6628	0.0256

## Data Availability

Not applicable.

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
