# Peer review of "Precise Modeling of Thermal and Strain Rate Effect on the Hardening Behavior of SiC/Al Composite"

_materials, 2022, doi:10.3390/ma15062000_

Round 1

Reviewer 1 Report

I have reviewed this paper earlier, wherein the authors implemented my suggestions. Unfortunately, The authors have now added ANN model. This addition is proving counterproductive.

The ANN is bound to improve the fit as it is a model with a large number of coefficients. Concurrently, the model losses the predictability for unknown data.

The authors may opt to remove the ANN ( as it is incomplete) or split experimental data into two sets, major and minor. The major set can be used to develop the model and the minor can be used to validate/check the predictability of the model. 

Author Response

Dear the reviewers and the editors,

Thank you very much for your time and efforts for the careful reviw of our manuscript and invalueable comments to improve the manuscript. Our replies to each comments are attached and the manuscript was revised accordingly.

We are looking forward to hearing you positive feedback at your earliest convenience.

Yours sincerely

Yanju Wang and Yanshan Lou

Reviewer 2 Report

This paper investigated the deformation behavior of the 15%SiC/Al 2009 composite under various temperatures and strain rates by tensile tests. The predicted results of the JC, ZA, MKH and MLH models and the BP network were compared with the  experimental true stress-plastic strain curves. In addition, it was reported that the ANN-based BP model is dramatically improve the predicting accuracy of the yield stress at different strains, strain  rates and temperatures.

The results of this paper are of great scientific value, especially the ANN-based BP model has been shown to be effective in the mechanical analysis of composite materials.

Please check that the following points do not affect the accuracy of the calculations, and if not, this paper should be accepted.

(1) Wasn't the SiC dissolved in the aluminum matrix?
If so, the various constants used in the calculations may be different from those of aluminum.

(2) Is it correct that the calculation does not distinguish between the elastic and plastic regions? 
The deformation mechanism is different in these regions, so please check to be sure.

Author Response

(The authors gave the same response as above.)

Reviewer 3 Report

The paper is good and nice presented. Unfortunately small correction need to be done. In revision your paper kindly have in mind the following:

  1. The abstract is a bit long. In my opinion firstly the authors should emphasize the novelty of this research and why the reader would be interested in this paper, because employing artificial neural network to characterize the flow behavior of the composite is, really, very interesting.
  2. In the Experimental L, LT and ST planes must be explained maybe using additional figure.
  3. In the figure 1 the scale bar is hardly visible.
  4. The caption of the figure 2 - Two-dimensional structure of uniaxial tension specimen (unit: mm)- may be simplified as - Uniaxial tension specimen (unit: mm).
  5. Referring Result chapter the authors should explain behavior of composite at low temperature (298K).
  6. As a general point of view the paper is too long. We should remark the meticulosity of the authors but reloading the same analysis for all four models may be too much for common reader. The information may be condensed. The figure 15 in my opinion is a good example. Anyway in the figure 15 b the BP network is hardly visible.
  7. In the Conclusions a short phrase with limitations and future work may be added and at point 3 the phrase:

Compared with the conventional models, the ANN-based BP model is shown to dramatically improve the predicting accuracy …

may be:

Compared with the conventional models, the ANN-based BP model is shown to highly improve the predicting accuracy

  1. Check again the reference format.

Author Response

(The authors gave the same response as above.)
